# Fuzzy Logic Modeling of Land Degradation in a Loess Plateau Watershed, China

**Ang Lu [1,2], Peng Tian [2,*], Xingmin Mu [1,3], Guangju Zhao [1,3] , Qingyu Feng [4], Jianying Guo [5] and Wenlong Xu [3]**

1   State Key Laboratory of Soil Erosion and Dryland Farming on the Loess Plateau, Institute of Soil and Water Conservation, Northwest A&F University, 26 Xinong Road, Yangling 712100, China
2   College of Natural Resources and Environment, Northwest A&F University, 3 Taicheng Road, Yangling 712100, China
3   Institute of Soil and Water Conservation, Chinese Academy of Sciences and Ministry of Water Resources, 26 Xinong Road, Yangling 712100, China
4   State Key Laboratory of Urban and Regional Ecology, Research Center for Eco-Environmental Sciences, Chinese Academy of Sciences, Beijing 100085, China
5   Institute of Water Resources for Pastoral Area of Ministry of Water Resources of China, 128 Daxue East Street, Hohhot 010020, China
*   Correspondence: pengtian@nwafu.edu.cn; Tel.: +86-29-8701-2411

**Abstract:** Various land degradation processes have led to land productivity reduction, food insecurity and ecosystem destruction. The Loess Plateau (LP) suffered from severe land degradation, such as vegetation degradation, soil erosion and desertification. This study assessed land degradation changes by considering different land degradation types including vegetation degradation, soil erosion, aridity, loss of soil organic carbon and desertification in the Huangfuchuan watershed of the northern LP. A comprehensive land degradation index (*LDI*) was developed by combining different degradation processes using the fuzzy logic modeling method. Our results showed significant land use transitions from bare land and sandy area to grass land and forest land from 1990 to 2018, which were consistent with an obvious increase in vegetation cover from 31.24% to 40.72%. The soil erosion rate predicted by the RUSLE model decreased by 51.95% during 1990–2018. The basin-average *LDI* decreased from 0.68 in 1990 to 0.51 in 2018, suggesting the great success of land degradation prevention in a fragile ecological environment region on the LP during the past decades. This study proposed an integrated framework for land degradation assessment in the high erodible area. The results can provide good references for the improvement of ecological environment in the future.

**Keywords:** land degradation index; fuzzy logic modeling; spatial and temporal distribution; Huangfuchuan watershed; Loess Plateau

## 1. Introduction

Land resources provide fundamental materials, such as food, fiber, and medicine, for human survival and development [1,2]. However, unreasonable land uses have led to critical land degradation in the past decades throughout the world [3], which threatens one-third of the global land area and affects more than 3 billion people, especially rural communities in poverty [4]. Land degradation results in land productivity reduction, population displacement, food insecurity, environment pollution and ecosystem destruction [5–7]. Evidence has been detected suggesting that land degradation is becoming an increasing threat, both to regional and global security in general [8,9].

Land degradation has become one of the most serious environmental problems, attracting great attention from both governments and global international researchers. Previous studies have demonstrated that land degradation involves multiple forms of environmental problems, requiring interdisciplinary and multidimensional investigation [10,11]. Significant efforts have been undertaken to assess the land degradation processes and extent

for various ecosystem types across different study sites. For example, Akinyemi et al. [12] assessed physical, chemical and biological degradation by using the Composite Land Degradation Index (CLDI) in an African agro-pastoral region. Yang et al. [2] simulated the spatiotemporal land degradation changes through analytic hierarchy processes with five indices. Masoudi et al. [11] developed a risk assessment of land degradation model (RALDE) considering various parameters including natural, human, and degradation trends. Turner et al. [13] reported six broad clusters of potential driving forces resulting in land degradation (such as climatic factors, technological factors, economic factors, political and institutional factors, demographic factors, and cultural factors) adapted from Geist and Lambin [14]. Prăvălie [10] reported that there currently were 17 land degradation pathways, i.e., aridity, landslides, loss of soil organic carbon, soil acidification, permafrost thawing, pollution, salinization, vegetation degradation, water and wind erosion, etc., which were active in various spatial scales. Among them, five land degradation processes, comprising water erosion, aridity, loss of soil organic carbon, salinization and vegetation degradation, were the dominant types of land degradation with reduced ecosystem service functions world widely.

In recent years, techniques of remote sensing (RS) and geographic information systems (GIS) have been widely applied to investigate land degradation processes and risks [15–18]. Several studies assessed land degradation with an individual process, such as vegetation degradation [19], soil erosion [20,21], and land desertification [22]. These research studies have facilitated the mechanism of land degradation at the regional and global scales. Many researchers also assessed comprehensive land degradation in different regions through different models [2,11,12]. However, it is difficult to reveal the land degradation processes and its spatiotemporal variation. The processes and mechanisms of land degradation vary by region and are complex [23,24]. Therefore, an integrated assessment of land degradation using various variables to represent key degradation processes could give more reasonable and accurate results for land degradation mitigation.

The most common spatial models used in GIS-based land degradation studies fall into deterministic models, such as Boolean overlay, and weighted linear combination [25]. However, these models are subject to deterministic structures and are not best compatible with multi-degradation processes. Compared to deterministic models, the fuzzy logic method could address the variability, imprecision, and ambiguity of the degradation processes [26], and has been used to assess land suitability of specific plants [26–28], soil quality index in terms of land degradation and desertification [29], and land subsidence risk [30]. Apparently, the fuzzy logic method is an efficient tool to qualitatively assess various land conditions, though most of them focused on a single land change process.

The Chinese Loess Plateau (LP) is characterized by severe land degradation, such as vegetation degradation, soil erosion and desertification [31,32]. The land degradation caused by soil erosion is the most serious issue in the LP. It is mainly distributed in the loess hilly area and plateau-gully area. In recent years, the LP has experienced significant changes to the land surface with implantations for numerous soil conservation practices and ecological restoration projects [33,34]. The "Grain for Green Program" (GFGP) initiated in the late 1990s was the largest ecological restoration project aiming to reduce soil loss, mitigate severe flood risk, and improve the livelihoods of the LP [35]. This project has successfully improved the vegetation cover leading to a visible "greening" trend in the LP [36]. Meanwhile, the soil erosion rate and sediment load to the Yellow River have been significantly reduced [37]. Considerable studies were undertaken to assess the change in soil erosion and vegetation cover on the LP [38–41]. However, the changes of various land degradation processes on the LP have been rarely investigated. Furthermore, limited studies have been undertaken to clarify the responses of land degradation to the implementation of soil-water conservation and ecological restoration. Therefore, the objectives of this study are to (1) comprehensively analyze the multidimensional land degradation processes (i.e., water erosion, loss of soil organic carbon, desertification, aridity and vegetation degradation) in the Huangfuchuan watershed of the northern LP; and (2) develop an

integrated framework to assess key land degradation processes by using the fuzzy logic method via combing remote sensing images, field survey and modelling data. The results could provide a scientific approach and reference for land degradation prevention on the LP and similar regions.

## 2. Materials and Methods

### 2.1. Study Area

The Huangfuchuan watershed is located in the northern LP (110°18′–111°12′E, 39°12′–39°54′N), with an area of 3246 km$^2$ (Figure 1). The Huangfuchuan river originated from southern inner Mongolia has a length of 137 km, and flows into the midstream of the Yellow River. The watershed is characterized by a semi-arid continental climate, with mean annual precipitation of 380–420 mm and annual temperature of 6.2–7.2 °C. The frequently occurring storms between June and September have led to severe soil erosion and substantial sediment yield, nearly 80% of which are concentrated in the wet season [42]. The deeply weathered coarse sandstone and the highly erodible loess resulted in high sediment yield in the watershed [43]. In particular, the bare deeply weathered sandstone covered with very sparse or no vegetation has extremely high sediment yield, and contributes approximately 70% of the coarse sediment in the watershed [44]. The Huangfuchuan watershed experiences multiple land degradation pathways due to its fragile ecological environment, though previous studies confirmed evident reduction in soil erosion and increase in vegetation cover [37].

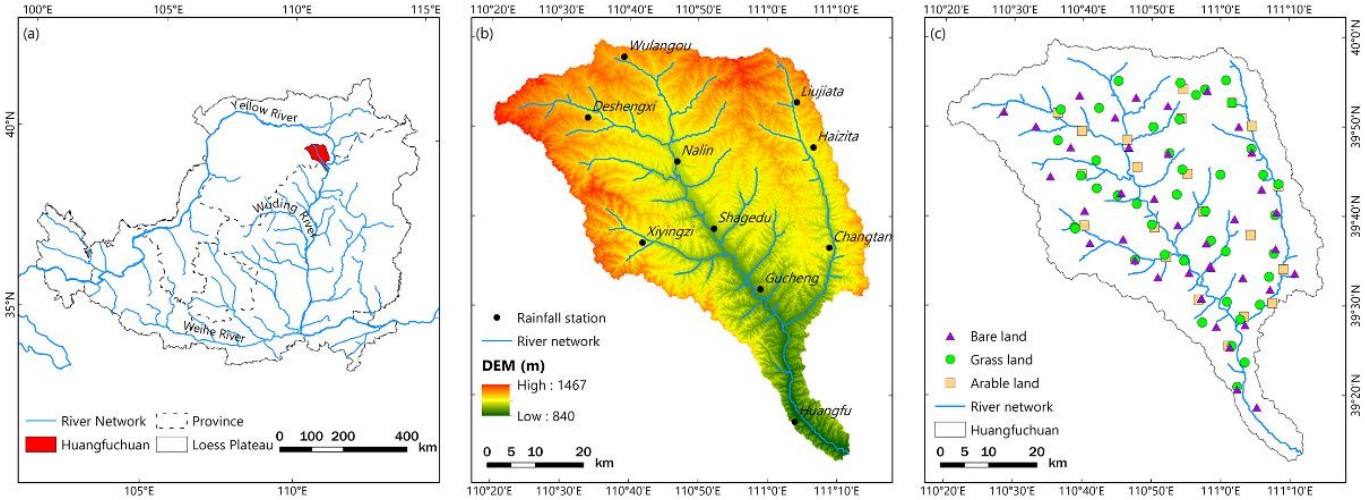

**Figure 1.** Location of the study area. (**a**) The Huangfuchuan watershed in the LP. (**b**) The rainfall stations in Huangfuchuan watershed. (**c**) The soil sampling points in different land uses in Huangfuchuan watershed.

### 2.2. Data Sources

We collected various data through interpretation of remote sensing images, field sampling, and geospatial analysis. The data consisted of meteorological variables, a digital elevation model (DEM), land use/cover, vegetation cover, and soil properties, which are listed in Table 1. Furthermore, these data were applied to assess land degradation via fuzzy logic modelling during different periods in the study area.

**Table 1.** Data used for land degradation assessment in the Huangfuchuan watershed.

| Data | Time | Resolution | Data Sources |
|---|---|---|---|
| DEM | - | 30 m | Geomatics Center of Shaanxi Province |
| Land cover | 1990 | 30 m | Landsat 5 TM, 22 August |
| | 2000 | 30 m | Landsat 5 TM, 23 July |
| | 2011 | 30 m | Landsat 5 TM, 10 September |
| | 2018 | 10 m | Resources and Environmental Sciences and Data Center, Chinese Academy of Sciences (Landsat 8 ETM) |
| NDVI | 1990 | 30 m | Landsat 5 TM, 13 August and 11 September 1989 |
| | 2000 | 30 m | Google Earth Engine |
| | 2011 | 30 m | Google Earth Engine |
| | 2018 | 30 m | Google Earth Engine |
| Soil properties | 2017 | - | Field Sampling |
| Climate | 1979–2018 | - | China National Climate Center |
| Precipitation | 1990–2019 | 10 stations | Yellow River Water Resources Commission |
| Terrace | 2018 | 30 m | Field investigation |

The daily meteorological data in the Huangfuchuan watershed were derived from the China meteorological forcing dataset (1990–2018), which can be downloaded in the China National Climate Center. Daily rainfall at 10 stations was acquired from the "Hydrological Yearbook of the Yellow River Basin (1990–2018)" (Figure 1b). The data quality, consistency and accuracy were checked by the agencies before their release.

Digital elevation model (DEM) with 30 m resolution was obtained from the Geomatics Center of Shaanxi Province and processed using 3D analysis in ArcGIS 10.7 (ESRI). The DEM was used to estimate soil erosion by making use of the topographical parameters of the watershed, e.g., the slope gradient and slope length, etc.

The data source of land use with different periods (1990, 2000, 2011, 2018) has been introduced in Zhao et al. [32] and Xu et al. [45]. The data were interpreted from TM or ETM images with a resolution of 30 m through supervised classification (Table 1), and had been verified through field survey to guarantee data quality. Seven land use types were identified: sandy land, forest land, arable land, urban area, grass land, water body, bare land. Among all periods, the dominant land use type was grass land, accounting for more than 70% of the watershed.

The annual normalized difference vegetation indexes (*NDVI*) in 2000, 2011, and 2018 (same as land use, 30 m) were estimated by Google Earth Engine (GEE) using the maximum synthesis method. Due to the lack of data on GEE, *NDVI* data in 1990 were calculated using Landsat 5 TM remote sensing image data (USGS, http://www.usgs.gov.vom (accessed on 20 May 2021)).

A total of 106 sampling points were selected to collect soil samples in the Huangfuchuan watershed (Figure 1c). The sampling points covered different land use and soil types, and were obtained in June 2017. The soil properties (particle size, soil organic carbon etc.) of the samples were measured to calculate soil erodibility factor (K) for soil erosion modelling.

All the spatial data were resampled to 100 m grid size using the nearest neighborhood method. These spatial data were used to estimate different land degradation maps.

### 2.3. Fuzzy Logic Modeling of Land Degradation Assessment System

An integrated land degradation index (*LDI*) was developed to consider multiple degradation processes based on the fuzzy logic modelling method (Figure 2). Firstly, several dominant land degradation processes were selected, i.e., vegetation degradation, soil erosion, aridity, loss of soil organic carbon and land desertification in the Huangfuchuan watershed. Secondly, each *LDI* was estimated or simulated by its corresponding models. Different land degradation maps were generated during the periods. Finally, land degradation maps were used as input layers to obtain the *LDI* using fuzzy logic theory with the procedure of fuzzification, fuzzy rule inference, and defuzzification (Figure 2). The *LDI*,

ranging from 0.0 to 1.0, denoted the degree of land degradation with a low value of no degradation and high value of severe degradation. The *LDI* maps were estimated by using the Arcpy in ArcGIS 10.7. More detailed procedures are described as below.

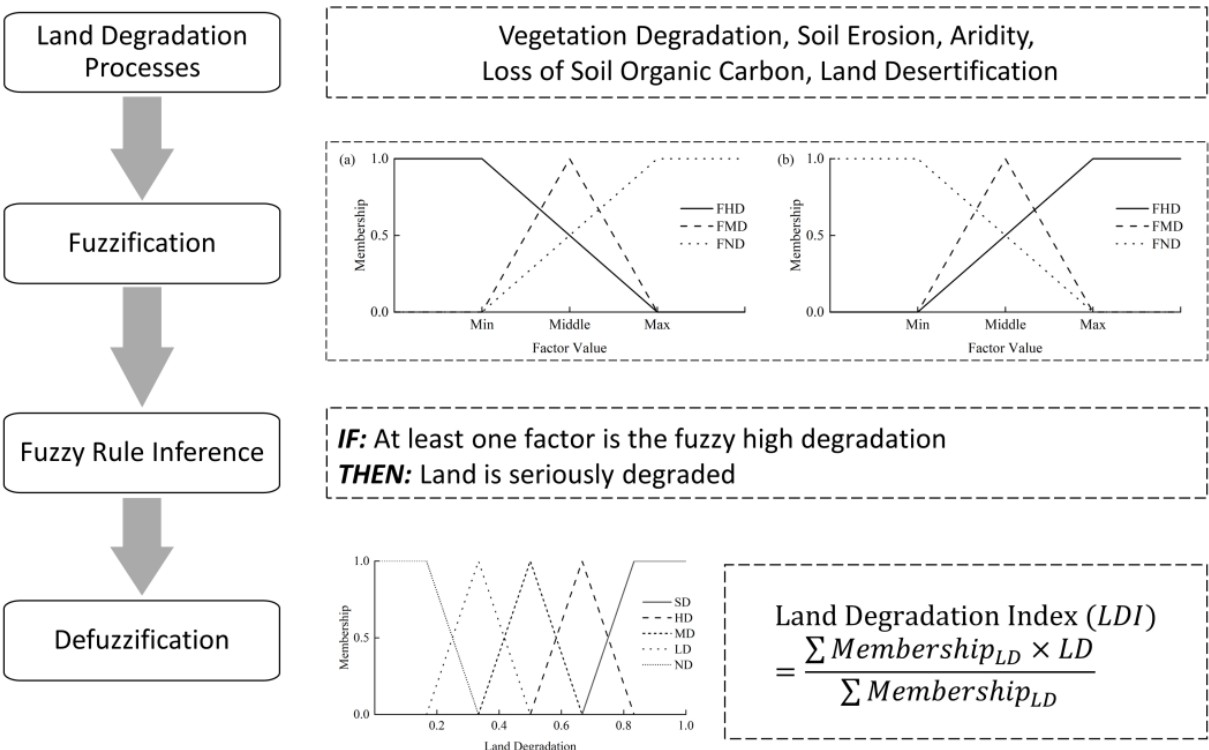

**Figure 2.** Flowchart of land degradation assessment based on the fuzzy logic modeling method. (**a**) The membership of vegetation degradation, aridity and loss of soil organic carbon. (**b**) The membership of soil erosion and desertification.

### 2.3.1. Dominant Land Degradation Processes

Land degradation is the result of a combination of natural factors (e.g., climate, soil, topography and vegetation) and human activities (e.g., farming, grazing and development and construction) [46]. Selecting dominant processes that can organically link the internal mechanism and external representation from many land degradation processes has become the key to establishing the land degradation assessment system. The selected land degradation processes should reflect the practical problems of the watershed, be easy to quantify and calculate, and have physical significance and representativeness. According to the previous research studies and field survey [10,13,16,31,47], we selected five dominant land degradation processes, including vegetation degradation, soil erosion, aridity, loss of soil organic carbon and land desertification to assess land degradation in the study area.

1. Vegetation degradation

Vegetation degradation is mainly manifested in the changes of vegetation types, vegetation structure and vegetation cover. Among them, the change of vegetation cover (*VC*) can most intuitively show the vegetation degradation. Significant correlation has been confirmed between vegetation cover and *NDVI* [48], and pixel dichotomy model is usually used to calculate vegetation cover. *VC* can be estimated by the linear difference ratio between vegetation index and soil index:

$$VC = \frac{NDVI - NDVI_{soil}}{NDVI_{veg} - NDVI_{soil}} \tag{1}$$

where $NDVI_{soil}$ and $NDVI_{veg}$, respectively, represent the $NDVI$ value of bare soil pixels and complete vegetation cover pixels. Here, we used the 98% and 2% percentile of the $NDVI$ raster values for $NDVI_{veg}$ and $NDVI_{soil}$ in the study area, respectively.

2. Soil erosion

Soil erosion is the most serious land degradation process in Huangfuchuan watershed. Soil erosion rate is an important indicator to measure soil erosion intensity. The soil erosion rate was estimated by the revised universal soil loss equation (RUSLE) [49], which considers various influencing factors (*R*, rainfall erosivity factor; *K*, soil erodibility factor; *LS*, slope length and gradient factor; *C*, vegetation and crop management factor; *P*, soil conservation factor).

The rainfall erosivity factor (*R*) was estimated by the approach proposed by Zhang et al. [50] using daily data at 10 stations in the Huangfuchuan watershed (Figure 1b). Field measured soil properties were used to estimate the *K* factor via the soil-erodibility nomograph method [51] (Figure 1c). The slope length (*L*) and the slope gradient (*S*) factors were estimated using the approach from Liu et al. [52]. The *C* factor was determined by combining both land use and vegetation cover [32,53] to conform to the erosion features in the watershed. The *P* factor was determined according to Zhao et al. [39] with related soil conservation measures.

3. Aridity

Aridity is an important form of land degradation, especially in arid and semi-arid regions, which is usually expressed by aridity index [10]. Aridity index (*AI*) is an indicator of the degree of available water, which is expressed by the ratio of water budget to heat balance. In arid and semi-arid regions, *AI* is closely related to climate change studies, and can be estimated using the ratio of actual evapotranspiration to precipitation:

$$AI = \frac{P}{ET_0} \tag{2}$$

where *P* is precipitation (mm), $ET_0$ is actual evapotranspiration (mm) estimated by the FAO−Penman−Monteith approach [54]. The inverse distance weighting interpolation method was used to interpolate the data of meteorological stations around the watershed to obtain the *AI* data.

4. Loss of soil organic carbon

The soil organic carbon (SOC) denotes one of the key soil attributes, which has great impacts on many physical and biochemical properties of soil [7,55]. SOC loss is one of the important and popular land degradation indicators [56]. The soil organic carbon content of the soil samples was measured by the potassium dichromate oxidation method [57], and the SOC map in the whole basin was obtained by the Kriging interpolation method. However, the temporal changes of SOC maps were not considered since the historical data was not available. Thus, an unchanged SOC map was used for land degradation assessment.

5. Desertification

Desertification index, namely the percentage of bare sand, was used to measure wind erosion and desertification. Regions with sandy land and bare land use types were considered as desertification regions, and desertification indexes of regions with other land use types were assigned as 0. In the desertification regions, considering the role of vegetation, the desertification index was estimated by subtracting *VC* from Equation (1) (Figure 3).

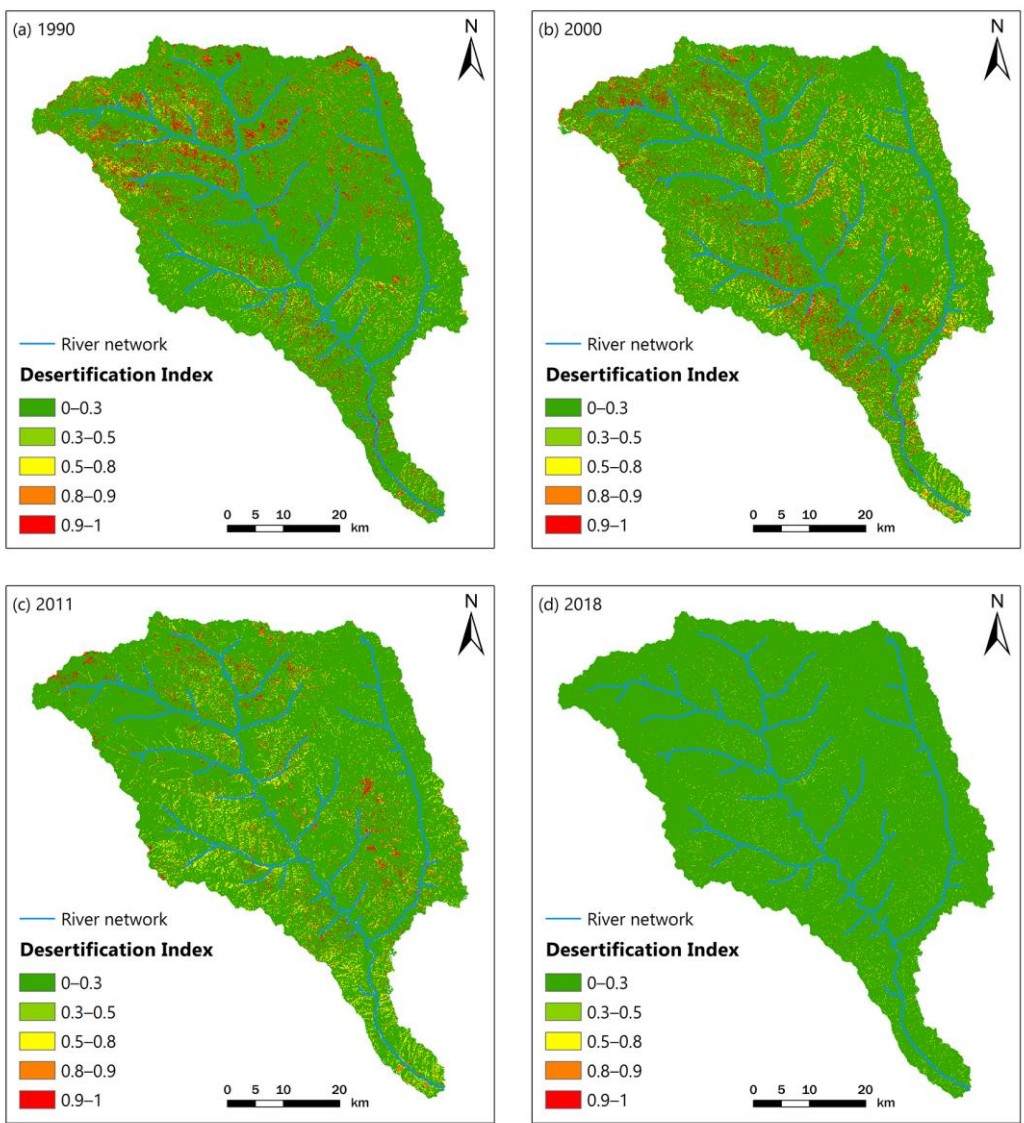

**Figure 3.** Desertification index changes in the Huangfuchuan watershed. (**a**) Desertification index map in 1990. (**b**) Desertification index map in 2000. (**c**) Desertification index map in 2011. (**d**) Desertification index map in 2018.

### 2.3.2. Fuzzification

Fuzzification is a process of converting the values of indices into fuzzy membership values through the fuzzy membership function, and unifying the values of all indices into the membership range of 0–1 for further analysis. Similar to the methods used by Joss et al. [36], three land degradation levels were established for each land degradation process: fuzzy high degradation (FHD), fuzzy moderate degradation (FMD) and fuzzy no degradation (FND).

Triangular fuzzy membership functions were used to calculate the membership values. Based on the relationship between indices and land degradation degree in different processes, two relative function forms were applied. The indices and land degradation degree of vegetation degradation, aridity and loss of soil organic carbon had the opposite relationship (Figure 4a). On the contrary, there were positive correlations between the indices and land degradation, such as soil erosion and desertification. The fuzzy membership function in Figure 4b was applied. The threshold of each fuzzy membership function is determined by the expert experiences, which are shown in Table 2.

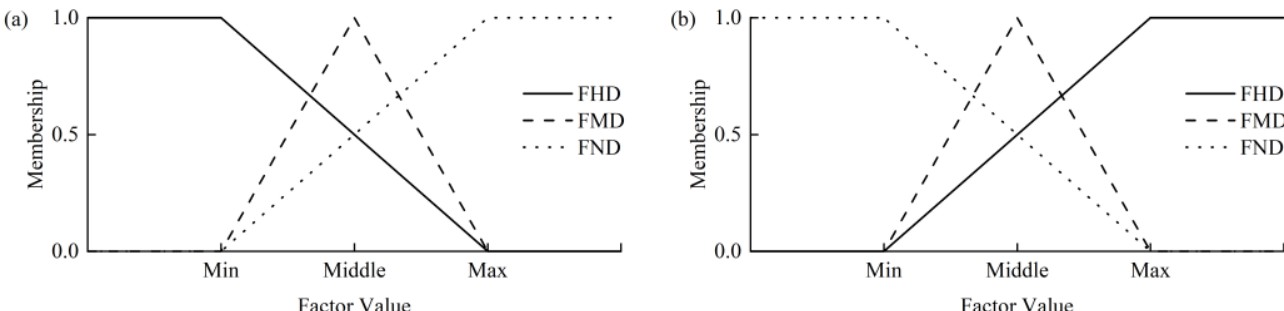

**Figure 4.** Fuzzy membership of factors. (**a**) The membership of vegetation degradation, aridity and loss of soil organic carbon. (**b**) The membership of soil erosion and desertification.

**Table 2.** Factors and their suitable ranges.

| Factors | Min | Middle | Max |
|---|---|---|---|
| Vegetation cover (%) | 10 | 30 | 50 |
| Soil erosion rate (t km$^{-2}$ a$^{-1}$) | 5000 | 10,000 | 15,000 |
| Aridity index | 0.03 | 0.34 | 0.65 |
| Soil organic carbon content (g/kg) | 5 | 10 | 15 |
| Desertification index | 0.3 | 0.55 | 0.8 |

### 2.3.3. Fuzzy Rule Inference

Fuzzy rule inference divides land degradation into five integrated degradation degrees based on all processes instead of only one process. The five degradation degrees include: serious degradation (SD), high degradation (HD), moderate degradation (MD), low degradation (LD) and no degradation (ND). The membership values denoted the land's degree of belonging to each of the five degradation degrees. The comprehensive fuzzy output value was estimated by the empirical IF-THEN reasoning rules including conditions (IF part) and conclusions (THEN part). Generally, the IF part is composed of one or more conditions connected through "And" or "Or" language [26]. For example, IF the vegetation cover is FND, soil erosion rate is FMD, aridity index is FMD, soil organic carbon content is FND, and desertification index is FND, THEN, the land is LD. All inference rules were designed based on professional knowledge. After determining the specific rules of application, the minimum–maximum (MIN–MAX) fuzzy rule inference method was used to execute the fuzzy rules inference [26].

### 2.3.4. Defuzzification

The land degradation index (*LDI*) was estimated by defuzzification by transforming the fuzzy membership values of five integrated degradation degrees from the fuzzy rule inference into a representative value [58]. Similar to the fuzzification process, the defuzzification process is realized by the fuzzy membership function. The fuzzy membership function diagram of *LDI* (Figure 5) was established to represent the membership value of each comprehensive degradation degree, by dividing the land degradation level from 0.0 to 1.0 into six sections on average. The weight of each comprehensive degradation level is determined by the center of maximum (COM) method [26]. The spatial distribution of *LDI* in each grid can be estimated by the weighted average calculation of *LDI* value.

$$LDI = \frac{\sum Membership_{LD} \times LD}{\sum Membership_{LD}} \qquad (3)$$

where *LD* is corresponding weight value of comprehensive degradation level, *Membership*$_{LD}$ is fuzzy membership value of comprehensive degradation level.

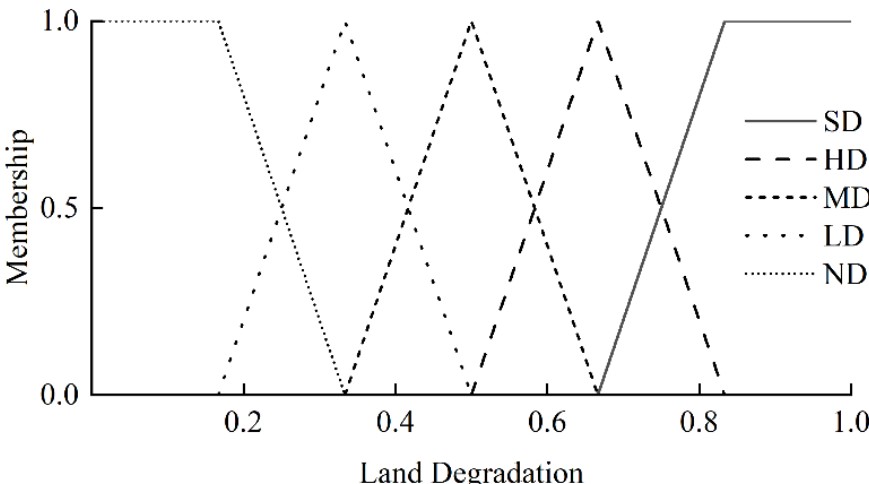

**Figure 5.** Membership of the different land degradation index degree.

## 3. Results

### 3.1. Changes in Vegetation Cover and Land Use

Spatial and temporal changes of vegetation cover (*VC*) can be clearly detected from 1990 to 2018 (Figure 6). Overall, we found high *VC* in the south and east, and low values in the northwestern watershed, which is consistent with the spatial distribution of precipitation and land use types. From 1990 to 2018, average *VC* of the watershed increased from 31.24% to 40.72%, and nearly 29.30% of the watershed had *VC* above 50% in 2018, indicating that the watershed vegetation cover improved greatly due to the afforestation started in 1999 by the Chinese government. An insignificant decrease in *VC* was detected in some small areas, which can be attributed to desertification, coal mining and urbanization.

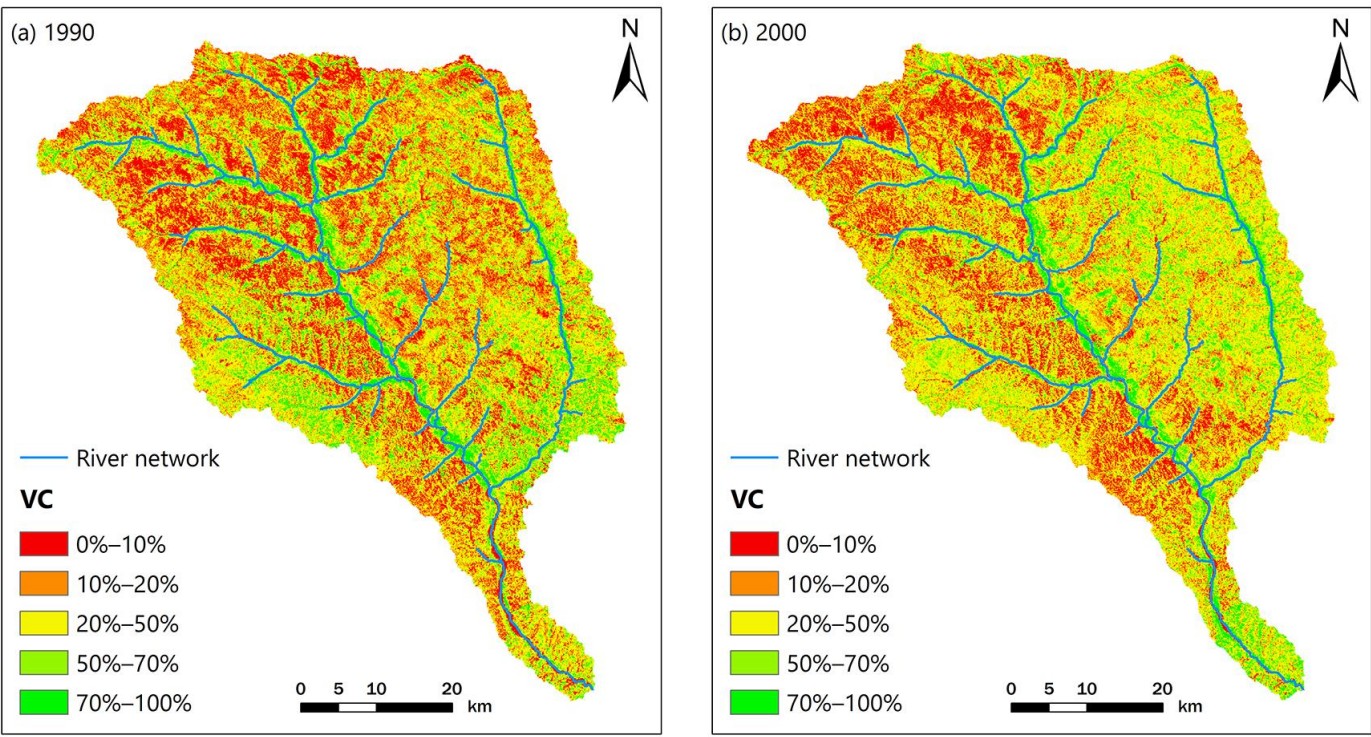

**Figure 6.** *Cont.*

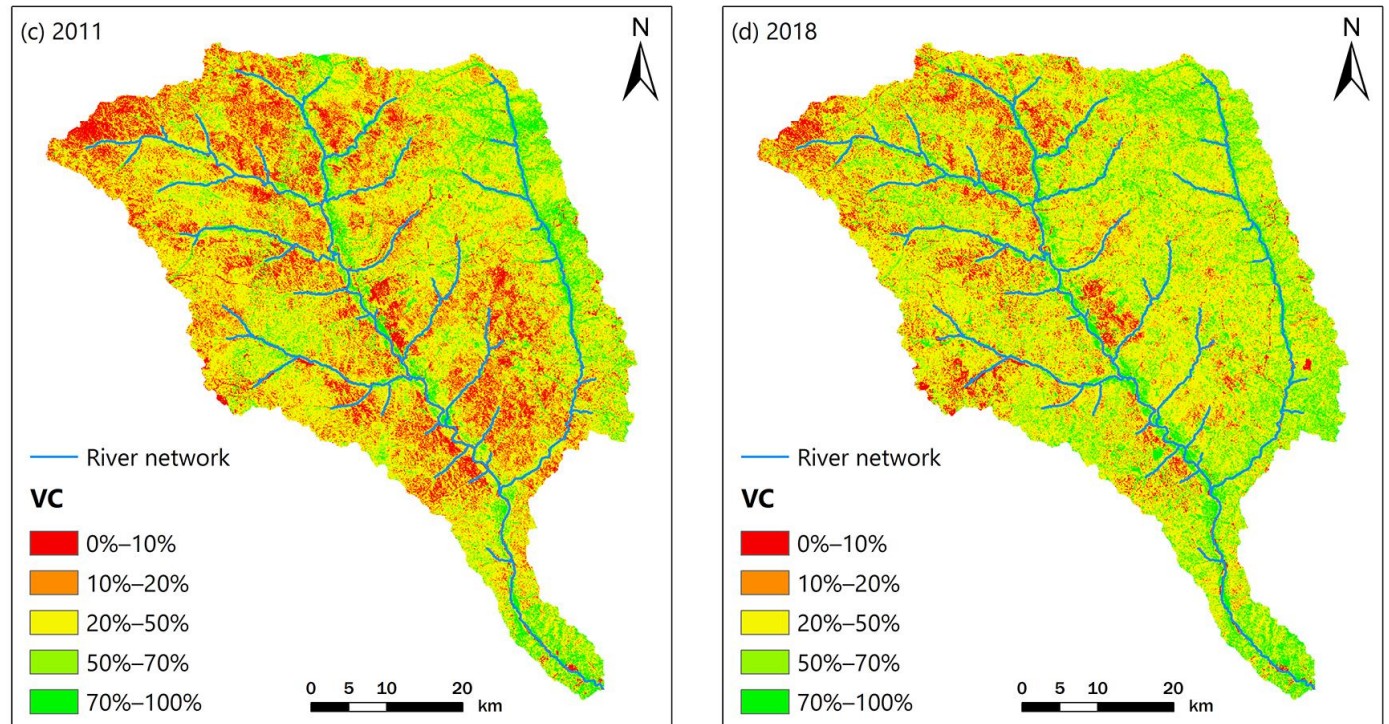

**Figure 6.** Vegetation cover (*VC*) changes in the Huangfuchuan watershed. (**a**) *VC* map in 1990. (**b**) *VC* map in 2000. (**c**) *VC* map in 2011. (**d**) *VC* map in 2018.

Figure 7 showed the spatial and temporal changes of land use between 1990 and 2018. Statistics indicate that the main land use type was grass land accounting for more than 70% of the Huangfuchuan watershed. Overall, arable land and sandy area were concentrated near the river, and urban area was mainly located in the middle reaches of the watershed. The bare land was mainly located in the steep slopes, and the forest land was scattered throughout the watershed. Changes in land use can be evidently examined from 1990 to 2018 (Figure 7). The forest land showed a great increase from 2.41% in 1990 to 8.37% in 2018. The arable land showed a slight decrease from 9.76% in 1990 to 4.37% in 2010, and then gently increased to 7.46% in 2018. The bare land was mainly located in the steep slopes, showing an obvious decrease from 8.27% in 1990 to 1.72% to 2018. Similarly, the sandy area decreased from 4.55% in 1990 to 0.56% in 2018. Between them, the bare land and sandy area covered a total area of 416.15 km$^2$ in 1990, and decreased to 74.01 km$^2$ in 2018. These changes showed that the policies, i.e., "returning farm land to grass land and forest land" in 1999 played an important role in land use changes. These trends were consistent with the obvious changes in the watershed's vegetation cover (Figure 6). In addition, the urban area showed a dramatic increase from 5.92 km$^2$ in 1990 to 76.06 km$^2$ in 2018, suggesting rapid urbanization and population increase during the past years.

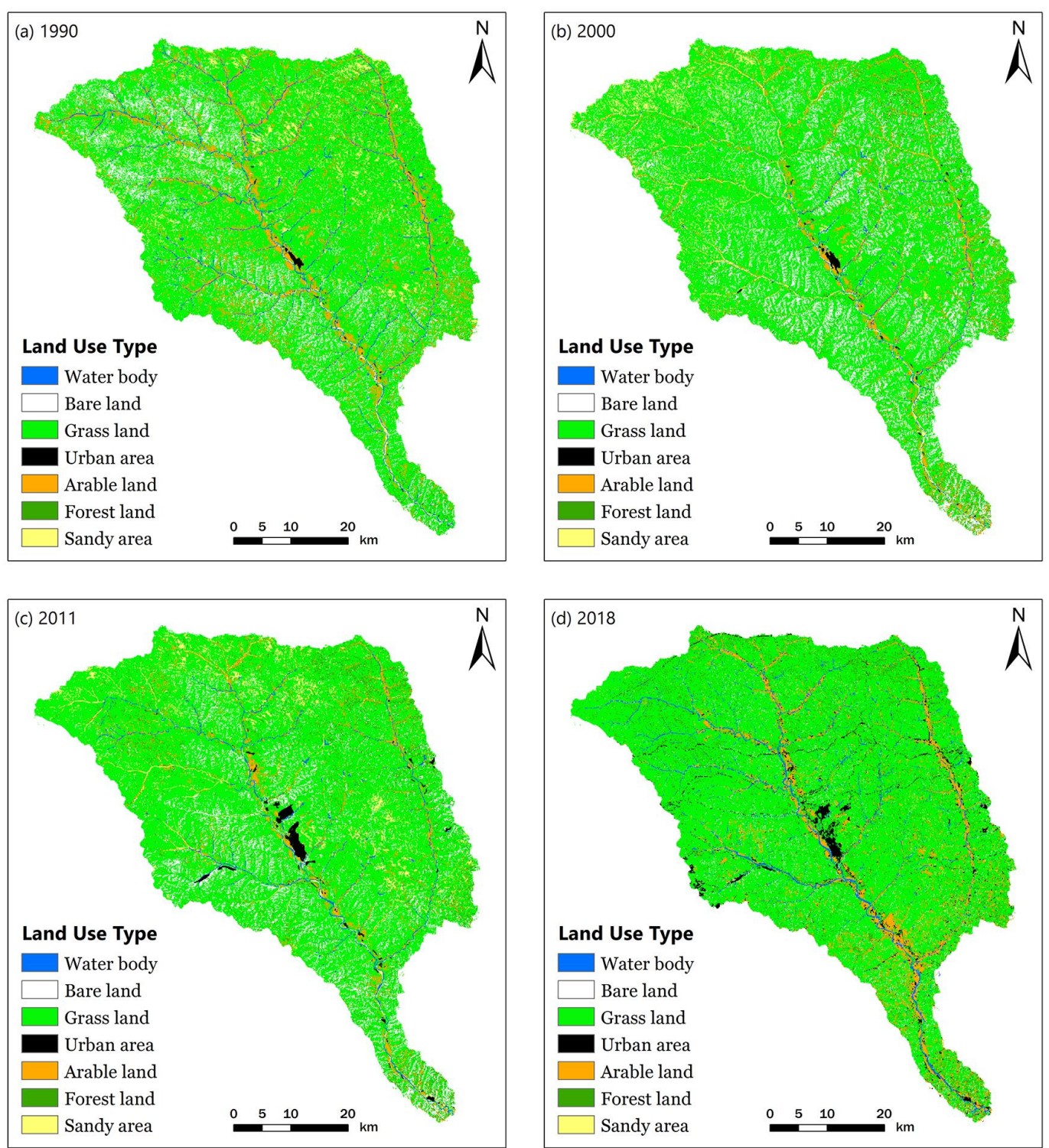

**Figure 7.** Land use changes in the Huangfuchuan watershed. (**a**) Land use map in 1990. (**b**) Land use map in 2000. (**c**) Land use map in 2011. (**d**) Land use map in 2018.

*3.2. Soil Erosion Rate Changes*

As the most serious and prominent land degradation process in the Huangfuchuan watershed, soil erosion was predicted by the empirical RUSLE model. According to the Chinese soil loss standard (SL 190-2007, made by Ministry of Water Resources of People of Republic China, 2007), the soil erosion rates can be divided into six classes: severe erosion (>15,000 t $km^{-2}a^{-1}$), very strong erosion (8000–15,000 t $km^{-2}a^{-1}$), strong

erosion (5000–8000 t km$^{-2}$a$^{-1}$), moderate erosion (2500–5000 t km$^{-2}$a$^{-1}$), slight erosion (1000–2500 t km$^{-2}$a$^{-1}$), and very slight erosion (0–1000 t km$^{-2}$a$^{-1}$). Figure 8 illustrates the spatial distribution of soil erosion during periods. Similar to vegetation cover and land use changes, average annual soil erosion rate reduced over 50% from 1990 to 2018.

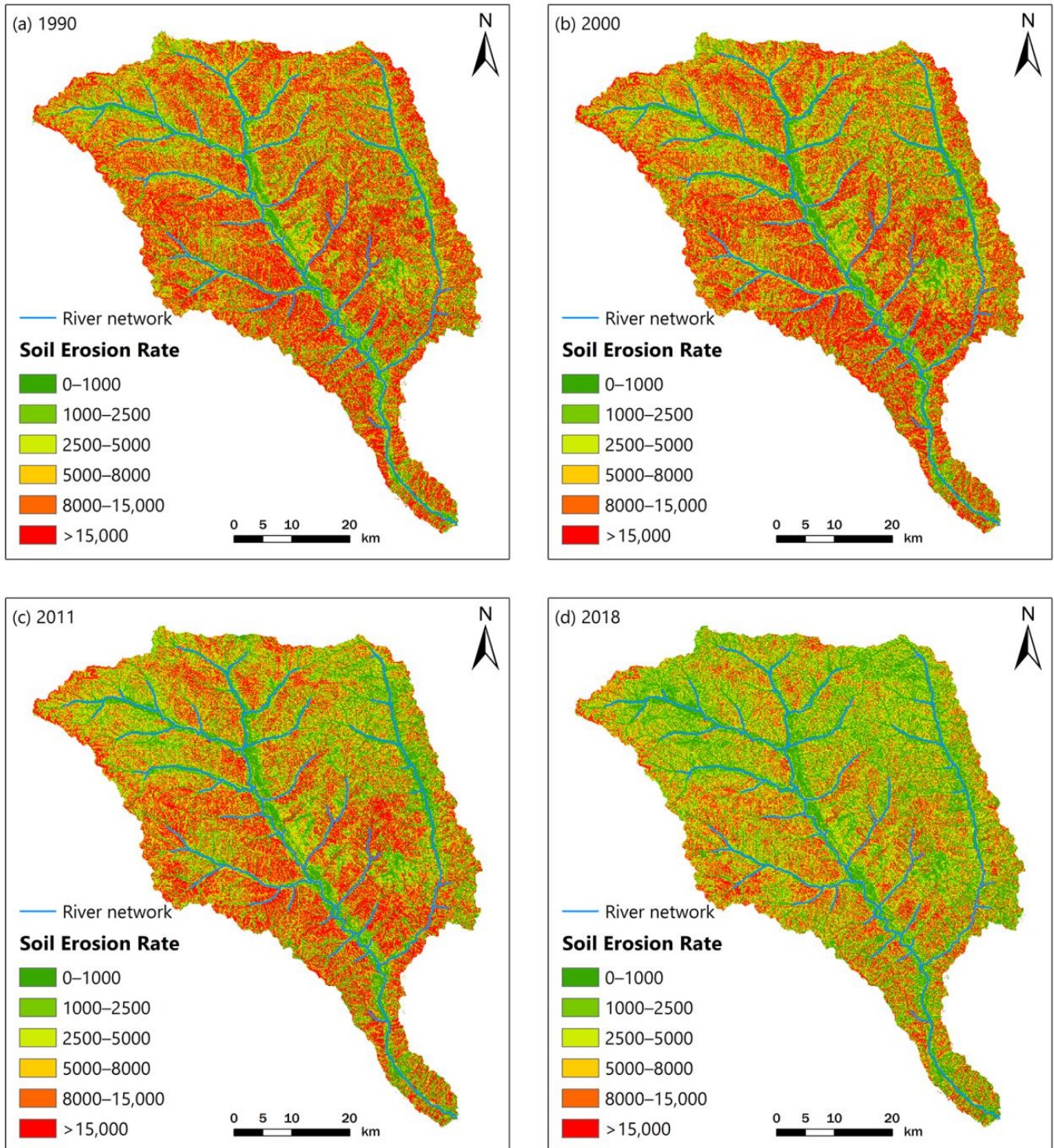

**Figure 8.** Soil erosion rate changes in the Huangfuchuan watershed (Unit: t km$^{-2}$a$^{-1}$). (**a**) Soil erosion rate map in 1990. (**b**) Soil erosion rate map in 2000. (**c**) Soil erosion rate map in 2011. (**d**) Soil erosion rate map in 2018.

In 1990, about 60% of the watershed was characterized by severe, very strong and strong soil erosion. Severe erosion area mainly distributed in the central and southern steep slopes of the watershed, with low vegetation cover and wide distribution of bare sandstone. By 2000, soil erosion rates had declined by 0.13%, while the trend was not significant. Afterwards, average annual soil erosion rate showed a rapidly decrease, indicating effective erosion control from various conservation measures i.e., "returning farm land to grass land and forest land" in 1999. The average soil erosion rate decreased from 12,869.05 in 2000 to 6191.37 t km$^{-2}$a$^{-1}$ in 2018. The area of severe erosion decreased significantly, from 909.85 in 1990 to 317.78 km$^2$ in 2018, with a decrease about 65%. Meanwhile, the very slight erosion area increased obviously with an increase of over 80%, from 500.86 to 913.10 km$^2$. By 2018, the very slight, slight and moderate erosion area accounted for 65.47% of the watershed, and was only 40.74% in 1990.

### 3.3. Land Degradation Changes

Figure 9 showed the spatial and temporal distribution of land degradation index (*LDI*) in the Huangfuchuan watershed. Overall, the *LDI* was classified into five grades according to the order from low to high: severe degradation (0.9–1), high degradation (0.8–0.9), moderate degradation (0.65–0.8), low degradation (0.4–0.65), and no degradation (0–0.4). Overall, areas with no degradation were concentrated in valleys and tablelands with flat terrain and high vegetation cover, while areas with high degradation were concentrated in tributary watershed with steep slopes and low vegetation cover. In 1990, the low degradation areas (<10%) were distributed in the channels in the eastern part of the watershed, and the severe degradation areas accounted for 42.01% of the watershed. By 2018, the low degradation areas with 48.02% whereas the severe degradation area reduced to 18.22%, and mainly located in the northwest. From 1990 to 2018, the average *LDI* has gradually decreased from 0.68 to 0.51, from moderate degradation to low degradation. Overall, land degradation was obviously transformed from high to low, especially near the vicinity of rivers and flat valleys.

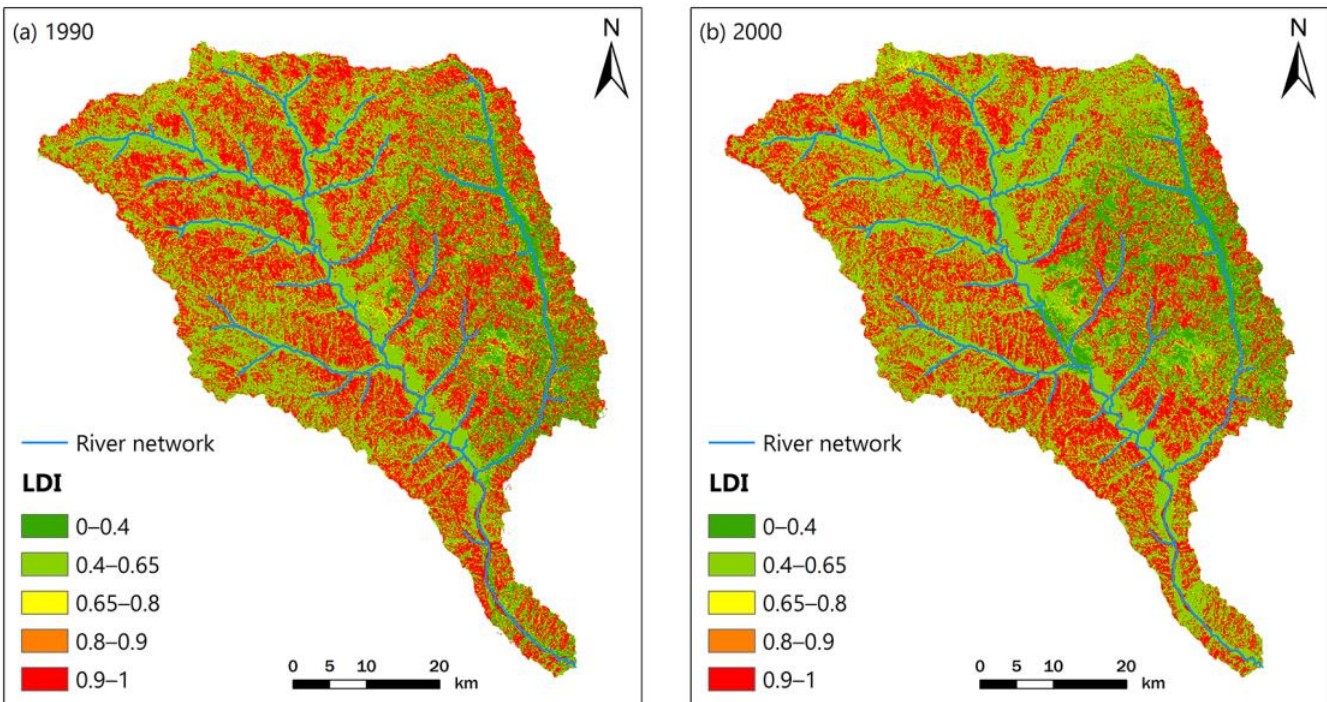

**Figure 9.** *Cont.*

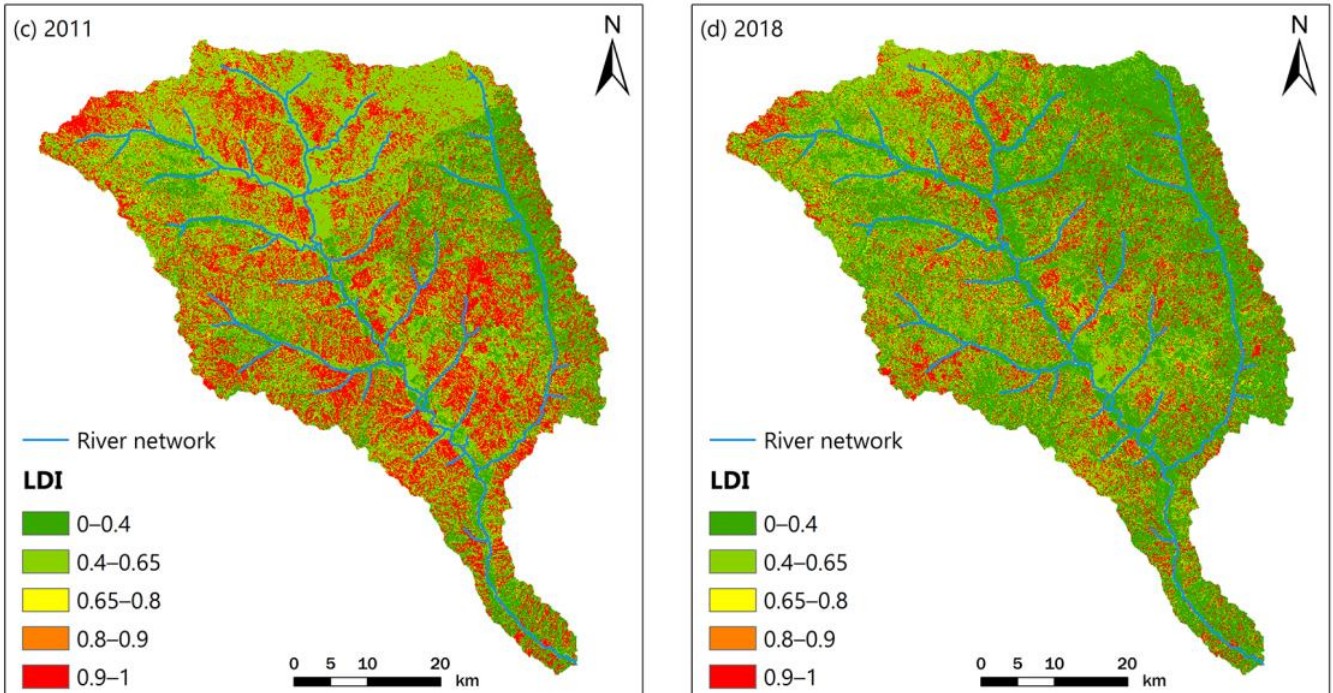

**Figure 9.** Land degradation index (*LDI*) changes in the Huangfuchuan watershed. (**a**) *LDI* map in 1990. (**b**) *LDI* map in 2000. (**c**) *LDI* map in 2011. (**d**) *LDI* map in 2018.

## 4. Discussion

### 4.1. Land Degradation Assessment

This study proposed a framework for land degradation assessment based on the fuzzy logic modelling method, which has been applied in many studies [26–30,59,60]. For fuzzy logic modelling, the selection of parameters must be noted firstly. Five dominant land degradation processes were selected using experts' empirical knowledge according to the local situations of the Huangfuchuan watershed. Vegetation degradation, soil erosion, aridity, loss of soil organic carbon and desertification were considered as the dominant land degradation processes in the study watershed.

Indicators representing each land degradation process set thresholds based on expert knowledge and past studies combined with filed survey. Based on the dominant position of sparse grassland in Huangfuchuan watershed, a relatively low vegetation cover compared with others [61,62] was selected as the maximum threshold representing the no degradation degree. Due to the serious soil erosion [37], we used 5000 instead of the commonly used $1000 \text{ t km}^{-2} \text{ a}^{-1}$ [63] as criteria to determine whether land degradation had taken place or not, in order to obtain higher parameter sensitivity. In some research studies, aridity index (*AI*) less than 0.65 is considered the beginning of land degradation [64]. By UNESCO [65], *AI* less than 0.03 means hyper-arid. These set the upper and lower limits of *AI* to express land degradation. Soil organic carbon thresholds were set at 15 g/kg and 5 g/kg based on the literature and watershed soil conditions [66,67]. Based on the importance of deeply weathered coarse sandstone and sandy land in the watershed natural conditions [43], the desertification index was set to relatively high thresholds. Although the method of selecting indices and their thresholds was subjective [60], field surveys were sufficient to confirm its rationality. However, indices and their thresholds may vary in other areas.

Due to the complexity of land degradation and the internal relationship between different land degradation processes [10], we applied the minimum–maximum fuzzy rule inference method rather than the weighted linear combination method to aggregate different land degradation processes [28]. The most important land degradation process was soil erosion in the study area, which was estimated by the RUSLE model. The watershed

has relatively complex geomorphology, and gully erosion and gravitational erosion are the main soil erosion types providing dominant sediment source [49]. Though detailed soil erosion processes cannot be estimated by the empirical RUSLE model, regional patterns and high erosion prone area could be detected with its widely application at continental and global scales [68–70]. In addition, the uncertainties of results caused by input maps could be reduced by the COM defuzzification method, because the small fluctuation of input maps does not alter the optimal compromise value of *LDI* [26].

### 4.2. Causes of Land Degradation Changes

Our results indicated that *LDI* decreased significantly in the Huangfuchuan watershed in the past decades. This was confirmed by the evident decline of watershed-averaged *LDI* from 1990 to 2018 [47,71]. Furthermore, *LDI* of different land use exhibited consistent trends, suggesting land improvement during the study period (Figure 10). *LDI* of grass land showed a stable decline from 0.67 in 1990 to 0.51 in 2018. Similarly, *LDI* of forest land and arable land decreased by approximately 0.11 and 0.08, respectively, which can be attributed to land use changes and the increase in vegetation cover (Figure 6). The area of forest land, grass land and arable land accounted for more than 80% of the watershed, thus their changes dominated land degradation processes. According to our analysis, vegetation restoration and soil and water conservation measures were responsible for vegetation improvement, decrease in soil erosion and desertification. These changes were the driving forces behind *LDI* changes. By contrast, the average *LDI* of bare land and sandy area, although declining, remained much higher than those of the other land uses. The high values of *LDI* resulted from low vegetation cover, SOC and severe soil erosion in these deeply weathered coarse sandstone areas. Thus, further conservation measures are needed to prevent land degradation and reduce soil erosion in the bare land with deeply weathered coarse sandstone and sandy area in the watershed.

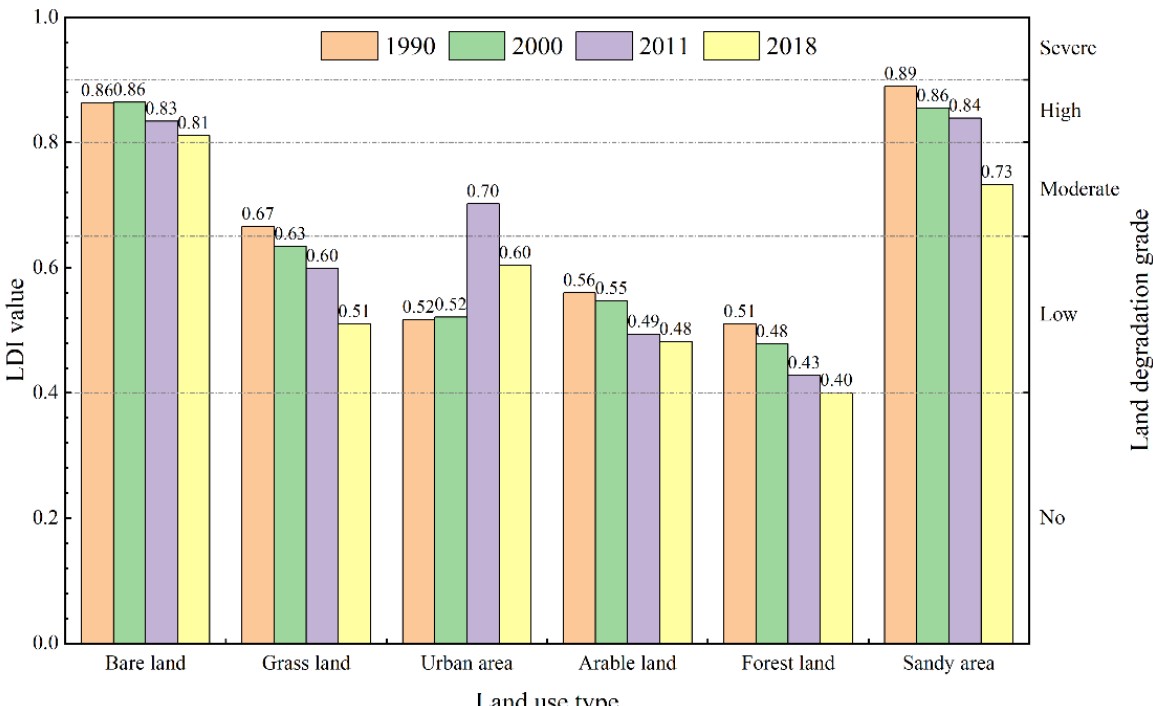

**Figure 10.** Land degradation index (*LDI*) changes with different land use types during different periods.

Apparently, vegetation restoration has been confirmed to be the main driving factor for the decrease in *LDI*. However, it has to be noted that water demand for vegetation in parts of the LP might have already approached sustainable water resource limits [72,73]. Water shortage and widely distributed dry soil layers require optimal plant species and a better

match of plant densities to the natural environment, rather than artificial trees in many parts of the region [35]. Though soil erosion was greatly reduced and vegetation cover showed significant improvement [32], the conflict between water resources and vegetation restoration existed in such an arid area.

Overall, the *LDI* values in the Huangfuchuan watershed of sandy land and bare land were still quite high (Figure 10), and vegetation restoration has proven to be an effective method to control land degradation. Therefore, considering the limitation of water resources on vegetation restoration, planting grass and shrubs together with hydraulic structure construction in the bare sandy land with deeply weathered coarse sandstone would be an important strategy to reduce land degradation in the Huangfuchuan watershed.

## 5. Conclusions

This study assessed spatial and temporal variation of land degradation in the Huangfuchuan watershed of the northern LP. A comprehensive *LDI* was developed with the fuzzy logic modelling method by considering different land degradation processes, i.e., vegetation degradation, soil erosion, aridity, loss of soil organic carbon and land desertification.

Overall, vegetation restoration led to an evident increase in vegetation cover from 31.24% in 1990 to 40.72% in 2018. An increase over 10% of grassland and forest land and a decrease in bare land and sandy area were detected due to the Grain for Green project initiated in 1999. Average annual soil erosion rate simulated by the RUSLE model exhibited more than 50% decrease from 1990 to 2018.

The integrated *LDI* showed an evident decrease from 0.68 in 1990 to 0.51 in 2018 in the Huangfuchuan watershed, suggesting that land condition and ecological environment had been improved due to various soil and water conservation projects during the past decades. A great success has been achieved in land degradation prevention and control in a fragile ecological environment region in the LP. This study proposed a valid framework to assess multidimensional land degradation processes in the high erodible watershed.

**Author Contributions:** Conceptualization, P.T. and G.Z.; methodology, A.L., W.X. and Q.F.; software, A.L. and W.X.; validation, Q.F.; investigation, A.L. and W.X.; resources, W.X.; data curation, W.X.; writing—original draft preparation, A.L. and P.T.; writing—review and editing, A.L., P.T., G.Z., X.M. and J.G.; visualization, A.L. and J.G.; supervision, P.T. and G.Z.; funding acquisition, P.T. and G.Z. All authors have read and agreed to the published version of the manuscript.

**Funding:** This research was funded by the National Natural Science Foundation of China (Grant Nos., U2243211, 42077076 and 42077075).

**Data Availability Statement:** Not applicable.

**Acknowledgments:** We thank the reviewers to provide valuable comments to improve the quality of this paper. Great appreciations are also given to all the data centers, which provided essential help for obtaining the datasets.

**Conflicts of Interest:** The authors declare no conflict of interest.

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
