# Peer review of "Fuzzy Logic Modeling of Land Degradation in a Loess Plateau Watershed, China"

_remotesensing, doi:10.3390/rs14194779_

Round 1

Reviewer 1 Report

This manuscript assessed land degradation changes by combining different land degradation processes using the fuzzy logic modeling method. Generally, this study was well organized, and the data analysis and model simulation were reasonable. And the results provided a good reference for the improvement of ecological environment in the future. I believe that this research is potentially a good contribution for land degradation control and a reference for the land degradation assessment in the high erodible area. However, some minor corrections should be further considered to improve the clarity of the manuscript. Therefore, I suggest a minor revision before accepting for publication in “Remote Sensing of Soil Erosion in Forest Area”.

1. Too many references, cited most recent and useful references.

2. Line 135, what about the data quality, consistency and accuracy?

3. Line144, explain what is the dominant land use, and their proportion.

4. In Figure 2, I find that the example of the third step “Fuzzy Rule Inference” is not consistent with the text.

5. Line 149, why don’t you estimate the NDVI in 1990 by GEE, but others were estimated by GEE?

6. Line 157, remove the sentence “Overall, different land degradation processes were assessed in 1990, 2000, 2011, 157 and 2018.” And explain how were these data processed for LDI estimation.

7. Table 2, how did you decide these threshold? Any references?

8. Lines 274-275 How Figure 5 was established should be described in more detail. The short terms need explanations (ISD, IHD, IND etc.)

9. Lines 297, 328 and 357 The word “changes” should be added to the figure name, such as “Vegetation coverage (VC) changes in the Huangfuchuan watershed”.

10. Figure 9, revise “LDI 1990” as “LDI in 1990”, and so on.

11. Line 413 The name of Figure 10 is not clear.

12. Lines 421-423 The logical relationship between the last sentence and the other sentences in this paragraph is not clear.

13. For any abbreviations, either in the text or the figures, explain them firstly and then use the short term.

Reviewer 2 Report

This manuscript assessed land degradation changes by combining multidimensional land degradation processes based on the fuzzy logic modeling method. This study proposed an integrated framework for land degradation assessment in the high erodible area, and the results could provide worthy references for the improvement of ecological environment in this region. The manuscript is readable, and well organized. The manuscript needs some improvements before publication in “Remote Sensing”. Detailed comments are as follows:

1.        Line 25, revise “during the study period” as “during 1990-2018”

2.        Line 26, revise “The average LDI” as “The basin-average LDI”

3.        Revise figure 1 to make it clearly.

4.        Lines 203-209 The estimation method of P factor is not introduced.

5.        Line 275, Figure 5 should be described.

6.        Throughout the manuscript, the “vegetation coverage” should be modified to “vegetation cover”.

7.        The authors may add river network in Figure 6, 8, and 9, as shown in Figure 3.

8.        Line 413, there is no obvious difference between the names of Figure 10 and Figure 9, which does not reflect the land use type.

9.        Lines 421-423 The last sentence “Thus, comprehensive assessment of land degradation processes should be taken into account for further rehabilitation of degraded land in the study area.”. This paragraph does not seem to have a direct causal relationship with the preceding sentences.

10.     Lines 29-30 “The results can provide good references for the improvement of ecological environment in the future.”. However, no recommendations were proposed for future land degradation prevention. If the authors gave more clear what is the dominant LD process, and then detailed comments, it would be valuable for local decision makers.

11.     In the references, there are 95 citations, better reduce 30% for non-related or non-useful ones.

Round 2

Reviewer 2 Report

Accepted